# Perception of four intellectual and developmental disabilities based on search engine and news portrayal

Lillian J. Droscha[1]*, Sophia Chung[2], Zoe Li-Khan[1], Ashley Scott[1], Eric Rubenstein[1]

1 Department of Epidemiology, School of Public Health, Boston University, Boston, Massachusetts, United States of America, 2 Department of Community Health, Tufts University, Medford, Massachusetts, United States of America

☯ These authors contributed equally to this work.

* ldroscha@bu.edu

## Abstract

### Background

For people with intellectual and developmental disabilities, other's perceptions of them based on their condition often begin before birth and go on to impact relationships, opportunities, and self perception across the life course. Search engine results and news media, which may portray these conditions stereotypically or in poor light, are often a key source in these perceptions. Our purpose was to understand how search engine results and available news media can shape perceptions on certain intellectual and developmental disabilities.

### Methods

We developed an online Likert-scale survey to measure differences in perceptions based off first available search engine results, images, and news headlines of four intellectual and developmental disabilities: cerebral palsy, Down syndrome, Prader-Willi syndrome, and Angelman syndrome. These four conditions were selected to compare less prevalent (Prader-Willi and Angelman) and more prevalent conditions (Down syndrome and cerebral palsy). Perception questions addressed general impression and aspects of the disability experience expected to be impacted by perception from others. We recruited via multiple social media platforms, flyers posted in the Boston area, and word of mouth to local communities and friends.

### Findings

229 individuals opened the survey, and 125 responses were used in analysis. Mean responses to Prader-Willi syndrome were significantly more negative than responses to cerebral palsy, Down syndrome, and Angelman syndrome across all variables. Responses to Angelman syndrome were also more negative than responses to Down syndrome. Significant differences between conditions found when treating the data as continuous were confirmed when treating the data as ordinal.

**Data Availability Statement:** The full raw data file, with all non-consenting and otherwise non-eligible

participants removed, can be found in the supplementary information labeled "S2 File.xlsx."

**Funding:** The author(s) received no specific funding for this work.

**Competing interests:** The authors have declared that no competing interests exist.

## Conclusion

Lesser-known intellectual and developmental disabilities, such as Prader-Willi syndrome and Angelman syndrome, are subject to more negative portrayal in media, leading to more negative perception, which may impact social opportunity and quality of life. Combined with our finding that the perception of Prader-Willi syndrome follows the ideals of the medical model of disability more closely than the social model, a need for social model of disability training and education for physicians and other medical providers is clear.

## Background

As of 2019, approximately 107.6 million people worldwide live with intellectual and developmental disabilities [1]. Intellectual and developmental disabilities are conditions that affect an individual's developmental trajectory during childhood and cause functional impairment. As a group of conditions, intellectual and developmental disabilities are heterogenous with varying effects on daily life and levels of awareness from the public. Four intellectual and developmental disabilities that highlight this heterogeneity are Prader-Willi syndrome, Angelman syndrome, Down syndrome, and cerebral palsy. Prader-Willi syndrome is a genetic disorder that can result in hyperphagia, low muscle tone and cognitive challenges. Angelman syndrome is a genetic disorder marked by developmental disability and nerve related symptoms, also related genetically to Prader-Willi syndrome [2]. Down syndrome is the most prevalent genetic cause of intellectual disability and is associated with complications in many body systems, due to the additional 21st chromosome [3]. Cerebral palsy is a congenital disorder that affects an individual's ability to move and maintain balance and posture [4].

Media representation has been shown to significantly impact the perception of people with intellectual and developmental disabilities for those who have little prior knowledge of intellectual and developmental disabilities [5]. Media can perpetuate common stereotypes of persons with intellectual and developmental disabilities, such as they are dangerous [6], victims or burdens [7], or inspirational figures miraculously overcoming disability [8]. In a study on the representation of Prader-Willi syndrome in major U.S. newspapers from 2000–2005, only ten out of thirty-two headlines about Prader-Willi syndrome were considered to be mostly positive or positive representations of persons with Prader-Willi syndrome. Articles considered as mostly negative or negative portrayals perpetuated stereotypes of persons with Prader-Willi syndrome as desperate, deviant, and out of control [9]. Samsel & Perepa found that media representation perpetuated harmful stereotypes that affected teachers' perception of disabilities [10]. The perception of intellectual and developmental disabilities has been shown to impact socialization [11], mental health [12], quality of healthcare received [13], and feelings regarding disability identity [14] for persons with intellectual and developmental disabilities. Although digital media has become a salient source of information used by many, no studies were found that examined the search engine representation of persons with intellectual and developmental disabilities. Additionally, there were no previously developed and validated measures of perception based on search engine results.

It is important to understand factors which contribute to perception of intellectual and developmental disabilities, such as media portrayal, as well as the differences in manifestations unique to varying disabilities. Our purpose was to understand how top search engine results and available news media impact the perception of four intellectual and developmental disabilities: cerebral palsy, Down syndrome, Prader-Willi syndrome, and Angelman syndrome.

## Methods

### Ethics statement

The Boston University Medical Campus Institutional Review Board approved this study. The approval and study number is H-44006. Written consent was obtained from anonymous survey participants through a question following a statement of study aims, risks and anonymity in the survey, by checking "yes, I consent to participate in this survey," or "no, I do not consent to participate in this survey." If participants did not consent, the survey automatically closed. No minors were included in this survey. We ensured this by beginning with a question with responses "yes, I am over 18," and "no, I am not over 18." If the participant was a minor, the survey automatically closed. Thus, there was no need for consent by proxy. All raw data collected from the survey is available in the S2 File.

### Study design

We developed an online survey to capture participants' perceptions of four different intellectual and developmental disabilities based on top search results in Google. We chose to focus on a pair of more well-known and prevalent conditions (Down syndrome and cerebral palsy) and a pair of lesser known and less prevalent conditions (Prader-Willi syndrome and Angelman syndrome) to understand potential differences in perception based on prevalence of the condition. We chose a quantitative survey design to analyze a larger sample size and provide an additional lens to existing qualitative literature assessing perceptions of disability.

### Search result collection

For Google search results, we used a new google account created for this project, with no prior searches on it, limiting Google's ability to algorithmically impact the results.

We performed a Google search and captured images of the visible screen for each condition:

'cerebral palsy', 'Down syndrome', 'Prader-Willi syndrome', and 'Angelman syndrome'. Initially, we conducted searches on the main Google search engine. We collected screenshots from the first search results page, including the top websites in the general search window, the Google featured snippet and the knowledge panel with information on the condition searched on the right-hand side. The Google featured snippet displays text from a popular, often governmental organization web page such as the CDC with information on what was searched. In addition, the featured snippet contains two to three top images. The knowledge panel, visible on the right-hand side of the page, contains detailed information on the symptoms, causes, and treatments of the condition that was searched, often with a light blue header. The knowledge panel also frequently contains an image representing the condition at the top, or a few images from the Google images tab at the bottom of the knowledge panel. After the general search, we moved to the Google images tab and collected images of the visible screen. We also collected the top five images from the Google Images search to provide a zoomed-in view of images an individual would see if they were to search the condition.

In addition, we searched the Google News tab for each condition and captured images of the first three resulting headlines, excluding scientific literature and aggregate articles. The format of the search term was as follows: "[condition] news." After searching the term, the first three qualifying links to articles were clicked on, and an image of the page viewed with the headline, sometimes photos, and the first part of the article was captured.

## Survey development

The questions included in the survey were designed to address a number of aspects of the experience of disability. As disability is recognized as a marginalized identity, it is understood to impact an individual from multiple angles. One way to conceptualize these angles is through the Social Ecological Model [15]. The Social Ecological Model frames the impacts of exposures as coming from four levels: the individual, relationship or interpersonal, community, and societal. The Social Ecological model can also be used as a framework for intervention, highlighting levels at which a change made would have a positive effect on the outcome of interest. To understand how perception of disability affects the experiences of individuals with intellectual and developmental disabilities at multiple levels of the Social Ecological Model, survey questions were designed to address these levels. Additionally, we were interested in exploring the general impression of the condition based on the images, as a general impression may have impact at multiple levels of the Social Ecological Model, from interpersonal relationships to policy-level decisions. We also designed questions based on three unique negative experiences of disabled people: other individuals finding their physical appearance or behaviors upsetting, being seen as incapable or infantilized, and being seen as inspirational. While being seen as inspirational may seem positive at first glance, it is often associated with a perceived physical or symbolic deficit that must be overcome [16]. Combined with the medical model of disability, being seen as inspirational places the burden of access on the individual, and thus neglects the societal and structural discrimination disabled people face [17].

For each condition, survey respondents viewed all screenshots of the google search, snippet, knowledge panel, news headlines and images. After viewing these, they were asked to respond to a series of questions in response to these images for that condition. Respondents were asked to answer the questions based solely on their perception of the images provided and to not use any prior knowledge they had of the disabilities. The order in which the conditions appeared in the survey was determined randomly.

A total of seven survey questions were created to evaluate the perception of search results. To address perception, respondents rated their general impression of the condition on a five step Likert scale ranging from strongly positive to strongly negative. Further questions addressed the three unique experiences of disabled people described above on a similar five step Likert scale ranging from strongly agree to strongly disagree. The three qualifiers used were "capable," "inspirational," and "upsetting". We chose to reverse the direction of the first qualifier, "capable," because we wanted to understand how capable the people in the images were perceived to be, rather than how incapable. The final three questions were created to address levels of the social-ecological model. These questions addressed the interpersonal, community and societal levels respectively. Survey questions are presented in Fig 1. A transcript of the entire survey, including photos used, can be found in the S1 File.

## Data collection

The survey was disseminated through multiple avenues to obtain a varied sample of respondents.

To maximize our sample size considering available resources, convenience sampling was used.

We posted the survey on social media, including LinkedIn, Reddit, Nextdoor, Instagram and Facebook. Flyers inviting the public to take the survey via a QR code were posted at local schools, shops, and community centers in the Boston area. Responses were collected between September 23 and October 31, 2023, with a target number of responses of 100. A total of 229 responses were collected. Responses generated from the survey's preview function used by

**Fig 1. Sample of survey questions.** Survey questions were repeated for each condition of interest, in place of "X".

study staff (N = 12), those in which the respondent did not consent (N = 4), those in which the responded was not older than 18 (N = 4), and those in which the respondent opened the survey but did not complete it (N = 105), were removed from the dataset. A total of 125 completed survey responses were used for data analysis.

## Data analysis

We treated the Likert data in both continuous and ordinal manners. Responses were coded as follows: strongly disagree as one, disagree as two, neutral as three, agree as four, and strongly agree as five. First, treating the data as continuous, we used a one-way ANOVA to analyze whether any of the conditions differed from any of the others. If there was a significant difference between any groups, we conducted pairwise comparisons of each condition to each of the others. We then used multiple-way ANOVA to assess impacts of demographic covariates. We chose to assess age, educational attainment, having children, and knowing someone with intellectual and developmental disabilities as covariates, as we hypothesized that these each might impact the effect of the condition being viewed on response to questions on the Likert scale. If any of these covariates were found to be significant, differences in responses by condition would be determined from the adjusted model. If no covariates were found to be significant, values would be determined from the unadjusted model.

For ordinal analysis we used a Kruskal-Wallis test to determine potential differences in the expected distribution of responses across the Likert scale and the data which were observed.

All data were analyzed using SAS Studio.

## Findings

### Description of sample

A total of 125 participants were included in our analytic sample. Participants had a mean age of 39 years with a minimum of 19 and a maximum of 77 years (Table 1). Most participants

were highly educated, with 55.2% having postgraduate degrees and 26.2% being college graduates. More than half of participants had children and 77.6% of participants knew someone with an intellectual and developmental disabilities.

## Statistically significant models indicate differences in response by condition

There were statistically significant differences in the mean response to at least one condition, compared to the others, in all seven question variables. These model-level significant differences found when treating the data as continuous were confirmed when treating the data as ordinal. None of the analyzed covariates were found to be statistically significant. All values presented above were obtained by the unadjusted model in which condition was the only predictor of response. Frequencies of each response, for each condition studied, can be found in Table 2.

## Prader-Willi syndrome is perceived more negatively than all three other conditions

Overall, when averaged, the Likert-scale responses were significantly lower when responding to Prader-Willi syndrome compared to the others for five variables indicated in Table 3. Compared to the other three conditions, the average Likert-scale responses were higher in Prader-Willi syndrome for the upsetting variable, indicated in Table 3. Graphical representations of the frequencies of each Likert-type response for each condition, and for each variable, are found in the S1–S7 Figs. A graphical representation of the mean response for each variable, grouped by condition, is also found in the S8 Fig. The frequencies of the "Agree" response or equivalent "Positive" response in the impression variable, were significantly lower for Prader-Willi syndrome compared to the other three conditions in the same five variables as in the ANOVA model. These significant comparisons of Prader-Willi syndrome to the other three conditions are marked in the "Agree" column of Table 2. Inversely, for the upsetting variable, the highest frequency of the agree response was when responding to Prader-Willi syndrome.

**Table 1. Demographics of survey participants.**

| Total Participants = 125 | Mean | Standard Deviation |
|---|---|---|
| Age | 39 | 14 |
| **Total Participants = 125** | **Frequency** | **Percent** |
| Education Level | | |
| Less than high school | 1 | 0.8 |
| High school graduate | 8 | 6.4 |
| Some college | 14 | 11.2 |
| College graduate | 33 | 26.4 |
| Post-graduate degree | 69 | 55.2 |
| Children | | |
| Yes | 67 | 53.6 |
| No | 58 | 46.4 |
| Know someone with IDD | | |
| Yes | 97 | 77.6 |
| No | 28 | 22.4 |

Description of the survey sample. 55% of our sample had obtained a post-graduate degree and 77.6% knew someone with an IDD.

**Table 2. Frequencies of each response by condition.**

| | Strongly Disagree (N, %) | Disagree (N, %) | Neutral (N, %) | Agree (N, %) | Strongly Agree (N, %) |
|---|---|---|---|---|---|
| **Cerebral palsy** | | | | | |
| Impression | 3 (2.4) | 25 (20.0) | 57 (45.6) | 37 (29.6) | 3 (2.4) |
| Capable | 4 (3.2) | 32 (25.6) | 27 (21.6) | 55 (44.0) | 7 (5.6) |
| Inspirational | 6 (4.8) | 21 (16.8) | 35 (28.0) | 51 (40.8) | 12 (9.6) |
| Upsetting | 19 (15.2) | 50 (40.0) | 38 (30.4) | 17 (13.6) | 1 (0.8) |
| Friends | 10 (8.0) | 58 (46.4) | 37 (29.6) | 17 (13.6) | 3 (2.4) |
| Community | 6 (1.6) | 35 (21.6) | 22 (28.8) | 51 (44.0) | 11 (4.0) |
| Quality of Life | 2 (1.6) | 12 (9.6) | 28 (19.3) | 61 (29.1) | 22 (40.0) |
| **Down syndrome** | | | | | |
| Impression | 3 (2.4) | 14 (11.2) | 60 (48.0) | 42 (33.6) | 6 (4.8) |
| Capable | 3 (2.4) | 22 (17.6) | 28 (22.4) | 61 (48.8) | 11 (8.8) |
| Inspirational | 7 (5.6) | 26 (20.8) | 38 (30.4) | 45 (36.0) | 9 (7.2) |
| Upsetting | 24 (19.2) | 53 (42.4) | 33 (26.4) | 13 (10.4) | 2 (1.6) |
| Friends | 6 (4.8) | 32 (25.6) | 44 (35.2) | 37 (29.6) | 6 (4.8) |
| Community | 3 (2.4) | 16 (12.8) | 25 (20.0) | 69 (55.2) | 12 (9.6) |
| Quality of Life | 2 (1.6) | 7 (5.6) | 24 (19.2) | 72 (57.6) | 20 (16.0) |
| **Prader-Willi syndrome** | | | | | |
| Impression | 19 (15.2) | 71 (56.8) | 33 (26.4) | [a]2 (1.6) | 0 (0.0) |
| Capable | 3 (2.4) | 39 (31.2) | 43 (34.4) | 37 (29.6) | 3 (2.4) |
| Inspirational | 24 (19.2) | 75 (60.0) | 24 (19.2) | [a]2 (1.6) | 0 (0.0) |
| Upsetting | 6 (4.8) | 21 (16.8) | 37 (29.6) | [b]50 (40.0) | 11 (8.8) |
| Friends | 17 (13.6) | 65 (52.0) | 37 (29.6) | [a]5 (4.0) | 1 (0.8) |
| Community | 4 (3.2) | 48 (38.4) | 37 (29.6) | [a]32 (25.6) | 4 (3.2) |
| Quality of Life | 7 (5.6) | 41 (32.8) | 47 (37.6) | [a]23 (18.4) | 7 (5.6) |
| **Angelman syndrome** | | | | | |
| Impression | 3 (2.4) | 25 (20.0) | 79 (63.2) | 18 (14.4) | 0 (0.0) |
| Capable | 6 (4.8) | 31 (24.8) | 44 (35.2) | 42 (33.6) | 2 (1.6) |
| Inspirational | 7 (5.6) | 41 (32.8) | 53 (42.4) | 22 (17.6) | 2 (1.6) |
| Upsetting | 11 (8.8) | 59 (47.2) | 43 (34.4) | 12 (9.6) | 0 (0.0) |
| Friends | 5 (4.0) | 30 (24.0) | 58 (46.4) | 29 (23.2) | 3 (2.4) |
| Community | 2 (1.6) | 27 (21.6) | 36 (28.8) | 55 (44.0) | 5 (4.0) |
| Quality of Life | 4 (3.2) | 15 (12.0) | 46 (36.8) | 54 (43.2) | 6 (4.8) |

Note: Headings represent the Likert scale responses for the capable variable through quality of life. For grammatical reasons, the Likert scale responses for the impression variable were Strongly Negative, Negative, Neutral, Positive, and Strongly Positive.

[a]Cases in which the frequency of "Agree" in Prader-Willi syndrome was significantly lower than all other three conditions.

[b]Cases in which the frequency of "Agree" in Prader-Willi syndrome was significantly higher than all other three conditions.

This comparison is also marked in Table 2. All statistically significant comparisons of Prader-Willi syndrome to each of the other three conditions found in the ANOVA model were also found in the Kruzkal-Wallis test, when treating the data as ordinal.

## Discussion

Perception of intellectual and developmental disabilities is an area with limited research available, despite its impact on interactions, self-perception, and opportunities for people with intellectual and developmental disabilities. Our purpose was to examine top search engine results and available news media to understand the differences in representation and

**Table 3. Mean responses to each question variable by condition.**

|  | Cerebral palsy | Down syndrome | Prader-Willi syndrome | Angelman syndrome |
|---|---|---|---|---|
| **Impression** | 3.10 | 3.27 | [a]2.14 | 2.90 |
| **Capable** | 3.23 | 3.44 | 2.98 | 3.02 |
| **Inspirational** | 3.34 | 3.18 | [a]2.03 | 2.77 |
| **Upsetting** | 2.45 | 2.33 | [b]3.31 | 2.45 |
| **Friends** | 2.56 | 3.04 | [a]2.26 | 2.96 |
| **Community** | 3.21 | 3.57 | [a]2.87 | 3.27 |
| **Quality of Life** | 3.71 | 3.81 | [a]2.86 | 3.34 |

Mean likert-scale response to each question variable, grouped by condition.

[a]Question variables with a mean response to Prader-Willi syndrome significantly lower than all other three conditions.

[b]Question variable with a mean response to Prader-Willi syndrome significantly higher than all other three conditions.

perception of four different Intellectual and Developmental Disabilities. We focused on four specific intellectual and developmental disabilities: cerebral palsy, Down syndrome, Prader-Willi syndrome, and Angelman syndrome. Our results suggest an overall more negative portrayal and perception of Prader-Willi syndrome, compared to Cerebral Palsy, Down syndrome, and Angelman syndrome based on information from search engines.

Based on our findings, search engine results from Angelman syndrome are perceived more negatively than Down syndrome. Coupled with the more negative perception of Prader-Willi syndrome, our findings indicate that less prevalent intellectual and developmental disabilities such as Prader-Willi syndrome and Angelman syndrome may be perceived more negatively when mainstream media and search engines are used as the source of information. To confirm this finding, our methods could be repeated using other intellectual and developmental disabilities at varying degrees of prevalence. The contradiction between the capable and inspirational responses among the four conditions shows that "inspirational" representations of disability do not necessarily portray individuals as being capable. The contradiction in these results captures the nuanced ways that disability is stigmatized through the narrative of people "overcoming" disability(16). It is important to note that the experience of ableism is complex and varied, and one experience should not be viewed as worse or better than another. For example, our results show that persons with Down syndrome are more likely to be viewed as inspirational, whereas persons with Prader-Willi syndrome are more likely to be viewed in a negative light. Both of these experiences have the potential to negatively impact self-perception, opportunities, and social interactions for people with intellectual and developmental disabilities.

Perception of intellectual and developmental disabilities impacts individuals with intellectual and developmental disabilities throughout the life course. First in the parent-child relationship, a parent's reaction and reconciliation of a diagnosis can have substantial impact on the child's upbringing, self-perception, and thus quality of life. In a study on family reactions and responses after a child's Down syndrome diagnosis, parents remembered the diagnosis with vividness, often having negative reactions such as saying it was "tearing them apart." Sometimes, the diagnosis was so emotionally difficult for the parent that a state of shock occurred [18]. It was also found that many of these parents turned to Google to find pictures, medical advice, and stories [18]. In a study of families of patients with rare diseases in Italy, it was found that over half the families experienced increased anxiety from web information [19]. The visceral reactions of these families can be attributed partly to the overall negative internet representation of Down syndrome and other intellectual and developmental disabilities. The impact of perception of intellectual and developmental disabilities on an individual's

quality of life extends beyond relationships to the parent(s) with other important social connections. Educators, Direct Support Providers, friends, siblings, and other relatives also likely rely on web information to understand intellectual and developmental disabilities. Thus, the stereotypes found in web information likely impact interactions with persons with intellectual and developmental disabilities, causing the person with IDD to experience ableism. Ableism, which leads to disparities in access to healthcare and outcomes through multiple pathways, must be considered as a major contributor to health disparities [20].

With the increased use of the Internet as a source of information on disability coupled with limited media representation of rare disorders, stigmatized portrayals may have a more significant effect on the perception of persons with rare disorders. The overrepresentation of medicalized portrayals of rare disorders may also reinforce the medical model of disability. All disabilities can be viewed through either a social or medical model, in which social models require a focus on societal problems, rather than an individual's problems [17]. Findings show that increased awareness of the social model and disability perspectives can improve physician care for patients and colleagues with disabilities [21]. Combined with our finding that the perception of Prader-Willi syndrome follows the ideals of the medical model of disability more closely than the social model, a need for social model of disability training and education for physicians and other medical providers is clear. This may be a potential remedy for the discrimination those with less prevalent disabilities, such as Prader-Willi syndrome and Angelman syndrome, face. Similarly, autism advocates and scientists have been pushing society to view autism "through the lens of *neurodiversity*, where autism is seen as one form of variation within a diversity of minds," rather than as a medical paradigm [22]. Applying this model to rarer disabilities, such as Prader-Willi syndrome, and understanding disability as social identity rather than a medical condition may help combat the common negative perception and representation of rarer disabilities, thus leading to better quality of life. Studies find that disability bias is quite prevalent in clinical settings and that pediatricians need to address ableism [19]. Moving forward, there is hope for a more inclusive and positive world for all types of disabilities.

Limitations of this study include a possible over-representation of those who are highly educated. Highly educated individuals might have a greater awareness of intellectual and developmental disabilities which may affect how they perceived the search engine results. Though we had an overrepresentation of highly educated individuals in our sample, education level was not found to be a statistically significant confounder on the relationship between the condition being responded to and the Likert scale response to the question. Additionally, search engine algorithms are individualized, and our study represented search results found when using a new Google account, at one point in time. It is possible that at a different time or location, or with an individual's personal Google account, search results would have differed. Finally, this study analyzed differences between only four intellectual and developmental disabilities. Intellectual and developmental disability is a broad category, and to understand the true differences in perception between well-known and rare conditions, as well as other defining characteristics of any given intellectual and developmental disability, examining other intellectual and developmental disabilities in this way would be important.

As a widely accessed forum of information, the Internet plays an essential role in reinforcing or redefining the public perception of persons with intellectual and developmental disabilities. This study reveals how the current state of Internet representation of persons with intellectual and developmental disabilities perpetuates certain social attitudes and stereotypes, including the perception of persons with intellectual and developmental disabilities as subjects of inspiration rather than capable individuals. Because disability is a deeply diverse identity, the experience of stigma also varies widely among disabled individuals. Representations of

Prader-Willi syndrome were more often viewed in a negative light, which reflects the more extreme stigma associated with Prader-Willi syndrome compared to other intellectual and developmental disabilities. The results of this study reiterate the pervasiveness of stigma in representations of intellectual and developmental disabilities and disparities within the diverse spectrum of disability.

## Supporting information

**S1 Fig. Impression distribution.** Frequencies of each response to the impression question for each condition.
(TIF)

**S2 Fig. Capable distribution.** Frequencies of each response to the capable question for each condition.
(TIF)

**S3 Fig. Inspirational distribution.** Frequencies of each response to the inspirational question for each condition.
(TIF)

**S4 Fig. Upsetting distribution.** Frequencies of each response to the upsetting question for each condition.
(TIF)

**S5 Fig. Friends distribution.** Frequencies of each response to the friends question for each condition.
(TIF)

**S6 Fig. Community distribution.** Frequencies of each response to the community question for each condition.
(TIF)

**S7 Fig. Quality of life distribution.** Frequencies of each response to the quality of life question for each condition.
(TIF)

**S8 Fig. Mean responses plot.** Plot of mean response to each question variable, grouped by condition.
(TIF)

**S1 File. Survey transcript.** The full survey transcript, with all questions asked of participants.
(DOCX)

**S2 File. Full survey raw dataset.** All relevant raw data obtained from the survey, with individual sample IDs and nonsense tracking variables removed.
(XLSX)

## Author Contributions

**Conceptualization:** Lillian J. Droscha, Sophia Chung, Zoe Li-Khan, Eric Rubenstein.

**Data curation:** Lillian J. Droscha, Sophia Chung, Zoe Li-Khan.

**Formal analysis:** Lillian J. Droscha.

**Investigation:** Lillian J. Droscha, Sophia Chung.

**Methodology:** Lillian J. Droscha.

**Project administration:** Lillian J. Droscha.

**Resources:** Lillian J. Droscha, Eric Rubenstein.

**Software:** Lillian J. Droscha.

**Supervision:** Lillian J. Droscha, Eric Rubenstein.

**Validation:** Lillian J. Droscha.

**Visualization:** Lillian J. Droscha.

**Writing – original draft:** Lillian J. Droscha, Sophia Chung, Zoe Li-Khan.

**Writing – review & editing:** Lillian J. Droscha, Sophia Chung, Zoe Li-Khan, Ashley Scott, Eric Rubenstein.

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
