## [Decision Letter · Decision Letter 0]

5 Nov 2024

PONE-D-24-07520Perception of Four Intellectual and Developmental Disabilities Based on Search Engine and News PortrayalPLOS ONE

Dear Dr. Droscha,

Thank you for submitting your manuscript to PLOS ONE. After careful consideration, we feel that it has merit but does not fully meet PLOS ONE’s publication criteria as it currently stands. Therefore, we invite you to submit a revised version of the manuscript that addresses the points raised during the review process.

We look forward to receiving your revised manuscript.

Kind regards,

Robert Didden

Academic Editor

PLOS ONE

Journal Requirements:

Reviewers' comments:

Reviewer's Responses to Questions

**Comments to the Author**

1. Is the manuscript technically sound, and do the data support the conclusions?

Reviewer #1: Yes

2. Has the statistical analysis been performed appropriately and rigorously? 

Reviewer #1: Yes

3. Have the authors made all data underlying the findings in their manuscript fully available?

Reviewer #1: Yes

4. Is the manuscript presented in an intelligible fashion and written in standard English?

Reviewer #1: Yes

5. Review Comments to the Author

Reviewer #1: This is a useful article on an under researched area

1. On page 11 the authors do nor explain why they have chosen the word "inspirational"? It is nor clear to me why this term was chosen

2. On page 12 how did the authors arrive at the question where only one question is measuring a negative reaction (upsetting) while the other questions are positive (e.g. doing activities, being capable etc.

3. On page 17 the authors state that "an overall more negative portrayal and perception of Prader-Willi

syndrome, compared to Cerebral Palsy, Down syndrome, and Angelman syndrome" On page 18 however they say that " search engine results from Angelman syndrome are perceived more negatively than Down syndrome". this seems contradictory. Perhaps the authors should avoid arbitrary classifications such as "common" and "uncommon" which appear a little arbitrary given that only 4 syndromes featured in this study making such generalisations less credible.

4. On page 19 the authors state "Findings show that increased awareness of the social model and disability perspectives can improve physician care for patients and colleagues with disabilities". This is an important statement and perhaps needs more emphasis.

6. PLOS authors have the option to publish the peer review history of their article (what does this mean?). If published, this will include your full peer review and any attached files.

Reviewer #1: **Yes: **Ashok Roy

---

## [Author Response · Author response to Decision Letter 0]

16 Dec 2024

Dear PLOS ONE Editors: 

We thank the reviewer for their thorough and constructive feedback. We have addressed their concerns and believe we have a stronger and more impactful paper. The summary is below. 

1. On page 11 the authors do nor explain why they have chosen the word "inspirational"? It is not clear to me why this term was chosen.

Thank you for this observation. We have addressed this point at the top of page 6 in the manuscript. In short, we chose to ask about perception of images as inspirational due to the common perception of disabled people as “inspirational,” even when doing simple, mundane tasks. This perception, while seemingly positive on the surface to a non-disabled person, feeds into the placing of the burden of access on the individual, as in the medical model of disability (20), by implying a physical or symbolic deficit that must be overcome (16). 

2. On page 12 how did the authors arrive at the question where only one question is measuring a negative reaction (upsetting) while the other questions are positive (e.g. doing activities, being capable etc.

We wanted to maximize impact of each question and minimize potential response time. Therefore, we opted to focus more on the positive words and use fewer negative words in our framing. We wanted to use a possible antithesis of the “inspirational” concept to understand if the presence of images as inspirational would yield a perception of images as “not upsetting,” or if these perceptions could be found in tandem. 

3. On page 17 the authors state that "an overall more negative portrayal and perception of Prader-Willi

syndrome, compared to Cerebral Palsy, Down syndrome, and Angelman syndrome" On page 18 however they say that " search engine results from Angelman syndrome are perceived more negatively than Down syndrome". this seems contradictory. Perhaps the authors should avoid arbitrary classifications such as "common" and "uncommon" which appear a little arbitrary given that only 4 syndromes featured in this study making such generalisations less credible.

Thank you for this point. Conditions like Prader-Willi syndrome and Angelman syndrome are classified as “rare diseases” by organizations like NIH whereas Down syndrome and CP are not. Nevertheless, our language is imprecise. We have changed common and uncommon to more and less prevalent. 

4. On page 19 the authors state "Findings show that increased awareness of the social model and disability perspectives can improve physician care for patients and colleagues with disabilities". This is an important statement and perhaps needs more emphasis.

We have added this to the abstract to stress the importance of this finding. 

Thank you again for your insightful comments. With these modifications, we believe our manuscript will contribute even more impactfully to the PLOS ONE community and appreciate the opportunity to resubmit. 

Best Regards, 

Lillian Droscha, MPH

---

## [Editor Report · Decision Letter 1]

18 Dec 2024

Perception of Four Intellectual and Developmental Disabilities Based on Search Engine and News Portrayal

PONE-D-24-07520R1

Dear Dr. Droscha,

We’re pleased to inform you that your manuscript has been judged scientifically suitable for publication and will be formally accepted for publication once it meets all outstanding technical requirements.

Kind regards,

Robert Didden

Academic Editor

PLOS ONE
---

## [Editor Report · Acceptance letter]

30 Jan 2025

PONE-D-24-07520R1 

PLOS ONE

Dear Dr. Droscha, 

I'm pleased to inform you that your manuscript has been deemed suitable for publication in PLOS ONE. Congratulations! Your manuscript is now being handed over to our production team.

Kind regards, 

on behalf of

Professor Robert Didden 

Academic Editor

PLOS ONE